# ON-DEVICE TRANSFER LEARNING BASED ON MIXED PRECISION PARTITIONING

## ABSTRACT

The application of machine learning is becoming more widespread, with a growing number of use cases. The development of centralized data training and the exponential growth of data generation raise significant privacy and security concerns. On-device training offers a solution by enhancing privacy and reducing the need for communication between the cloud and the device. Furthermore, on-device transfer learning (TL) can leverage the knowledge gained from pre-trained models, hence, accelerating the training process. However, backpropagation, especially in embedded systems, requires more memory than running inference, which becomes a challenge for devices with limited resources. This paper aims to improve the efficiency and performance of on-device TL. We propose an open source mixed-precision partitioning framework that identifies optimal partitioning layers for retraining, combining quantized and bfloat16 layers to enhance performance and energy efficiency. Our approach is validated through experiments on ResNet-18 and SqueezeNetV1.1 models using Flowers-102, STL-10, and OxfordIIITPet datasets. The partitioned mixed-precision model is able to transfer the knowledge from the pre-trained model to new datasets without losing accuracy compared to the baseline bfloat16 model. These results illustrate the potential for resource-constrained devices to perform TL locally.

## 1 INTRODUCTION

Machine Learning (ML) in the past decade has been applied in various fields from healthcare (Javaid et al., 2022) to autonomous driving (Bachute & Subhedar, 2021), due to its well-known ability to derive patterns and make predictions from vast amounts of data. Traditional ML approaches often involve centralizing data in cloud servers in order to train a model. The following factors, such as the centralized training approach, the growth of generated data, and the continued adoption of ML algorithms, could pose critical concerns about the privacy and security of user data (Xu et al., 2021).

Addressing these privacy concerns is one of the reasons to use Transfer Learning (TL). This technique allows models to leverage the knowledge of the pre-trained models from another domain (Pan & Yang, 2010). TL is also applicable when limited data is available, reducing the need for extensive data collection. Many research papers have explored the potential of TL, for example, in the autonomous driving (Chen et al., 2024), and robotics (Zhu et al., 2023). However, to fully implement TL solutions into real-world scenarios, it is essential to bring the training process closer to the data source. This can be achieved through on-device training, where models are trained directly on devices in the deployed environment (Zhu et al., 2022).

On-device training offers several advantages, including improved data privacy, real-time model updates, and reduced latency in cloud-to-device communication, which is very important for an autonomous driving use case. Despite these benefits, performing training on a device is a challenging task due to the limited computational resources and memory constraints of embedded devices (Dhar et al., 2019), which are required for the inference and backpropagation on device. To overcome the limitations of hardware, common approach in TL is to freeze weights and biases of the feature extractor layers and only retrain the classifier. This technique enables the network adaptation to new data with less computational resources, but at the same time leads to the accuracy degradation of the adapted model. Moreover, current studies rarely address the problem of the partitioning point selec-

tion, before which all layers of the network are frozen. Hence, our paper fills this gap by answering the following research question:

- How can we identify a partitioning layer to freeze the preceding layers and retrain the subsequent ones in order to successfully and efficiently train on a device?

The contribution of the paper is as follows.

- We introduce an open source framework for mixed-precision partitioning for on-device TL.
- We present a new algorithm for the partitioning layer identification based on layer robustness analysis.
- We verified that the partitioned model, consisting of quantized and bfloat16 layers, can perform as well as a full bfloat16 model on new datasets.

Additionally, we made the code publicly available.

## 2 MOTIVATION

The influential paper of Yosinski et al. (2014) proved the possibility of Deep Neural Networks (DNNs) to transfer the learned features from one dataset to another. One of the main results demonstrated in that paper was the performance degradation of the model when only the top layers were retrained. As they concluded, the closer we get to the final layer, the less a model can relearn for a new dataset. Despite of their contribution, many scientific works, e.g. Chiang et al. (2023), still split models between the feature extraction and the classification layers during transfer learning. Table 1 shows the accuracy of the ResNet-18 model by splitting at the three feature extraction layers (first, penultimate, and the last one) and at the classification layer. After splitting, the upper layers were retrained on three datasets (more details in section 5). The model weights were initially pre-trained on the ImageNet dataset. As expected, splitting a model even one layer before the classifier significantly improves the model performance. Hence, our first motivation is partitioning the model before the classifier will increase the accuracy.

The mentioned work of Chiang et al. (2023) targeted a challenging task - transfer learning on embedded devices, such as NVIDIA Jetson Nano and Raspberry Pi 4. Due to the limited memory and computational resources of these devices, the backward pass computation should be highly optimized to achieve a lower memory footprint as well as lower latency of the forward and backward passes. It is thus apparent that the bottom layers cannot be considered viable candidates for use as a partitioning point in order to enable on-device transfer learning in such embedded systems. Moreover, the reduction in the number of layers undergoing retraining will result in enhanced memory efficiency with regard to backpropagation. As a result, this serves as a second key motivation for our work.

Finally, the work of Xiao et al. (2023) demonstrates the significant memory reduction by using `int8` precision instead of `fp16`. As stated, quantization is an effective method for reducing the model size and accelerating inference. Other works, such as Rossi et al. (2022), also showed the increased efficiency and performance of using `integer` rather than single-precision floating-point format for the presented Internet-of-Thing endnode system on chip. The `bfloat16` format seems to be the best trade-off between the training performance of a DNN and energy efficiency. As stated by Norrie et al. (2021), `bfloat16` works seamlessly for almost all ML training, while reducing hardware and energy costs. They estimated that `bfloat16` has approximately a $1.5\,\mathrm{x}$ energy advantage over the IEEE 16-bit float for the more recent $7\,\mathrm{nm}$ processors. This leads to our third key motivation, that the combination of `integer` and `bfloat16` will significantly increase performance and energy efficiency.

## 3 MIXED-PRECISION PARTITIONING FOR ON-DEVICE TRAINING

The optimization of on-device training for the purposes of improving model accuracy while reducing model size requires the identification of a beneficial trade-off. This trade-off must balance the opposing principles mentioned above. This section presents a framework that employs an optimized

Table 1: Test accuracy of the ResNet-18 model by splitting at the feature extraction and classification layers and retraining on three datasets.

| Datasets | Feature Extractor | | | Classifier |
|---|---|---|---|---|
| | First layer | Penultimate layer | Last layer | |
| Flowers-102 | 88.7 % | 87.8 % | 86.1 % | **78.3 %** |
| STL-10 | 94.7 % | 94.7 % | 94.5 % | **91.9 %** |
| OxfordIIITPet | 88.7 % | 87.8 % | 86.1 % | **78.3 %** |

mixed-precision partitioning methodology to enable energy-efficient on-device TL in embedded systems.

In the field of partitioning, a multitude of approaches have been published that address the problem of finding a beneficial trade-off for splitting the inference (Peccia & Bringmann, 2024). However, for the purposes of this work, we are only able to draw upon a limited number of these methodologies, as they typically do not investigate the impact on the model accuracy. In order to identify a partitioning scheme that enables on-device TL while maintaining high accuracy, it is necessary to conduct an in-depth analysis of the impact of each layer in the forward pass. Nevertheless, executing this procedure for each upcoming TL iteration would result in a significant computational overhead. Consequently, we propose a methodology that employs the pre-trained model for this analysis.

## 3.1 PRELIMINARIES

Before proceeding to the problem description and our approach, it is first necessary to formally define a DNN as well as a function to further quantize weights and activations. A DNN can be described as a graph comprising nodes and edges, representing layers and their respective connections. The objective of our methodology is to achieve a good trade-off between energy efficiency and on-device training performance in edge devices. Consequently, we assume that weights and activations are already provided in `bfloat16` number representation. Accordingly, a layer of a DNN is defined as follows:

**Definition 1** *A layer $l$ is a layer of a DNN with $bfloat16$ computational precision.*

As previously stated, embedded systems are constrained in terms of available memory and offer less performance than GPU-based HPC platforms due to their limited size and power consumption. Consequently, further quantization of individual layers to q-bit integers is beneficial in order to account for these limitations during deployment. For this purpose, a corresponding function is used, which is defined as follows:

**Definition 2** *A function $Q(l, q) = l^q$ is quantization of layer $l$ with q-bit integer computational precision.*

The implementation of our framework employs the use of ONNX as the input format of DNNs, which offers the benefit of inherent representation as a graph. This layer graph serves as the foundation for subsequent operations and explorations.

## 3.2 TOPOLOGICAL ORDERING

In modern DNNs, parallel branches or skip connections are utilised to address the issue of vanishing gradients during training. However, this architectural feature also has implications for the partitioning, as some layers in the layer graph receive input from multiple sources. Consequently, the search space for partitioning becomes significantly larger than that of a purely sequential DNN, due to the existence of numerous potential topological orderings for such models.

Based on the definition of Cormen et al. (2022), a topological sort is a linear ordering of the nodes in a Directed Acyclic Graph (DAG). Non-recurrent DNNs are acyclic and can therefore be represented

as a DAG, with the nodes representing associated layers. Based on this, we define the topological ordering of a DNN as follows:

**Definition 3** *A topological ordering of a DNN comprising a set $L$ of $m$ layers is a consistent enumeration of these layers and is given by $\varphi : L \rightarrow \{1, \ldots, m\}$ such that*

$$\forall l_1, l_2 \in L : \ \varphi(l_1) < \varphi(l_2) \Rightarrow l_1 \text{ is executed before } l_2,$$
$$\forall l_1, l_2 \in L, \ \ l_1 \neq l_2 \Rightarrow \varphi(l_1) \neq \varphi(l_2).$$

Consequently, in order to evaluate the robustness of each layer, it is first necessary to identify a valid topological ordering. In our framework, we use the Python library NetworkX provided by Hagberg et al. (2008) to derive a linear ordered set of layers for the subsequent exploration.

### 3.3 Layer Robustness Exploration

Based on the topological ordering of a DNN, we define the problem of finding an advantageous partitioning for efficient on-device training as follows. First, we generalize the definition of layerwise partitioning as proposed by Kreß et al. (2024). A partitioning point marks the first layer after the partitioning of the network:

**Definition 4** *A partitioning point is a layer $l_s \in L$ with $s \in \{1, \ldots, |L|\}$ in a DNN consisting of a set $P$ of sequentially executed partitions, such that*

$$\psi : \varphi \rightarrow \{1, \ldots, |P|\}, \ \ \forall l_1, l_2 \in L : \ \varphi(l_1) < \varphi(l_2) \Rightarrow \psi(\varphi(l_1)) \leq \psi(\varphi(l_2)),$$
$$\psi(\varphi(l_{s-1})) \neq \psi(\varphi(l_s))$$

In general, we assume that the sensitivity factor of each layer indicates its impact on the model's accuracy. This allows us to reduce the search space for partitioning to $\mathcal{O}(N)$, where $N$ is the number of layers. To conduct the in-depth analysis of the impact of each layer in the forward pass, the current state of the art primarily calculates the sensitivity of each individual parameter (Dash et al., 2022). Nevertheless, this approach entails a considerable runtime overhead, with the analysis requiring approximately an hour per eigenvector on two NVIDIA GTX1080 Ti GPUs for a ResNet-18. In contrast, we use the robustness of each layer to quantization as an effective and expedient indicator for identifying sensitive layers. Given the vast number of potential integer precision combinations within the search space, two simplifications based on characteristics of typical hardware architectures are applied in our framework to further reduce the runtime. Firstly, the activations and weights are quantized to the same integer precision. Secondly, only the relevant integer computational precision that can be implemented in the system is selected, i.e. 4-, 6-, 8-, and 16-bit integer. Remaining combinations are efficiently explored with the NSGA-II (Deb et al., 2002), similar to the methodology proposed by Hotfilter et al. (2023). As a result, the exploration algorithm can be defined as follows.

**Definition 5** *The exploration algorithm is an automated procedure that operates on*

1. *a DNN described by $L$ and $\varphi$,*

2. *a set of quantization functions $Q(l, q)$, where $q \in \{4, 6, 8, 16\}$, and*

3. *an accuracy threshold $a_{th}$*

*to find Pareto-optimal quantization schemes $s \in S$ of the DNN that provide an accuracy $a \geq a_{th}$.*

As an initial population, we generate 32 samples containing only quantizations of the two largest integer bit widths, i.e. 8- and 16-bit integer, to achieve fast convergence. Subsequently, these are evaluated in terms of accuracy and the sum of layer bit widths, after which a new generation is derived based on simulated binary crossover (SBX) and polynomial mutation (PM). In total, the framework assesses 20 generations to identify non-dominated solutions, as described in Definition 5. To accelerate the search process, we iteratively increase the number of validation samples over the generations, which are used to determine the accuracy, dismissing unpromising solutions early on. While the algorithm tries to maximize the number of layers quantized to low bit integer precision, it tries to maximize the accuracy. Hence, the multi-objective optimization can be defined as follows:

**Definition 6** *The goal of the multi-objective optimization is to find the Pareto front $S$ such that the number of integer quantized layers and the top-1 accuracy are maximized while the q-bit integer computational precision is minimized.*

## 3.4 MIXED-PRECISION PARTITIONING

The robustness analysis may yield multiple non-dominated solutions, depending on the DNN. Consequently, the framework must ultimately select a partitioning scheme that optimizes the trade-off for on-device TL. According to the results presented in Table 1, we remove the classification layers from the exploration. The selection of a point is typically driven by the specific requirements of the application domain. In certain scenarios, a higher degree of accuracy loss may be tolerated to enable significantly more energy-efficient on-device training. This is represented in the framework by a user-defined parameter, $\delta$, which denotes the maximum loss of accuracy compared to the non-dominated quantization scheme with the best accuracy $a_{best}$ found. As a result, the framework seeks a non-dominated solution $s \in S$ that offers a low $q$-bit integer computational precision while maintaining an accuracy $a \geq a_{best} - \delta$. This can be defined as a minimization problem, as follows:

**Definition 7** *The minimization problem for a set $S$ of quantization schemes is given as*

$$\underset{S}{\text{minimize}} \quad \sum_{i=0}^{|L|} q_i$$
$$\text{subject to} \quad a_i \geq a_{best} - \delta$$

*with the set of layers $L$ and the $q_i$-bit integer computational precision of a layer $l_i \in L$.*

For the following experiments we will use $\delta = 0.01$. So we allow a maximum loss of accuracy of 1 %. Once the partitioning layer is obtained by the algorithm, the DNN mapping can be formulated as follows:

**Definition 8** *The output of the framework is defined by*

1. *a set $\Omega \subset L$ that contains all layers $l \in \{l_1^q, \ldots, l_{s-1}^q\}$ (bottom layers) mapped to an accelerator with computational precision $q$, and*

2. *a set $\Theta \subset L$ that contains all layers $l \in \{l_s, \ldots, l_{|L|}\}$ (upper layers) mapped to an accelerator for training.*

As a result of our framework, the identified mixed-precision partitioning scheme can be implemented in embedded systems for on-device TL. The bottom layers $\Omega$ of a DNN, before the partitioning layer, are quantized and can be deployed on a lower bit-width accelerator. These layers can be thought of as the inference of a model. In contrast, the upper layers $\Theta$ are represented as `bfloat16` and can be deployed on another accelerator to adapt the model for a new dataset locally on the device.

## 4 EXPERIMENTAL SETUP

In this section, we present the used models, datasets, and our step-by-step experiment to prove the identified partitioning point for training of DNNs in embedded systems.

### 4.1 MODEL AND DATASET PREPARATIONS

The main idea of TL is to utilize pre-trained models on large datasets to derive learned features, and then apply them to improve the learning performance on a new dataset. In our case, we used image classification as a TL task. The ResNet-18 (He et al., 2016) and SqueezeNetV1.1 (Iandola et al., 2016) are the well-known models for image classification. In our experiments, the ImageNet dataset (Deng et al., 2009) was used as the large dataset, on which the models were pre-trained. In order to demonstrate our approach on the TL example, we used three additional image classification datasets, i.e. the Flowers-102 (Nilsback & Zisserman, 2008), the STL10 (Coates et al., 2011), and the OxfordIIITPet (Parkhi et al., 2012).

## 4.2 Experimental procedure

In our experimental procedure, we used the datasets and models mentioned above. In order to keep this work transparent, comparable, and reproducible for all interested researchers, we also provide additional details of the training setup. The details, such as learning rates, number of epochs, etc., can be found in Appendix A. The framework itself is omitted for blind review, but will be made publicly available.

The goal of this experiment is to demonstrate that the partitioning layer identified by the presented algorithm fulfills the two primary conditions: it represents the maximum number of quantized layers prior to partitioning with the highest possible accuracy of the model. In other words, the goal is to identify the uppermost layer to start retraining a model without losing accuracy. As a consequence, these conditions can enable energy-efficient training in embedded devices. This was achieved through the experiment, which consists of three main steps, shown as rows in Figure 1.

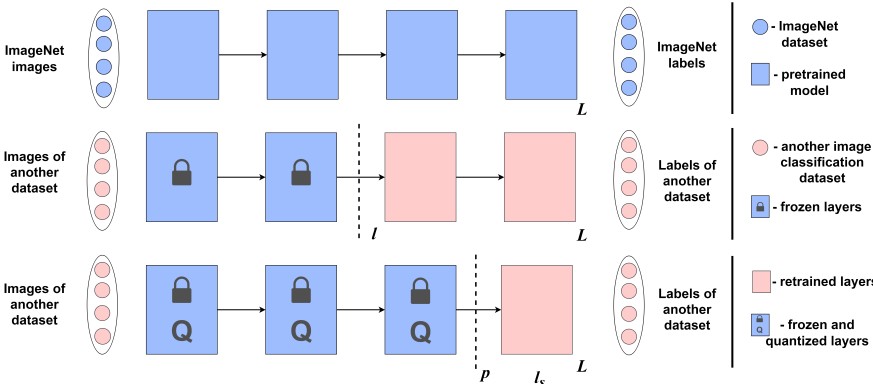

Figure 1: The main experimental procedure based on three main steps. The pre-trained model on the Imagenet dataset is shown in the *first row* (*blue*), the *second row* demonstrates the TL of the model to another dataset (*red*). The *third* depicts our final goal, i.e., the partitioned model with the frozen and quantized layers before the partitioning $P$, and `bfloat16` subsequent layers.

Each rectangle represents a simplified version of a layer in the DNN. The first row demonstrates the given pre-trained models on the ImageNet dataset (*blue*). The second row shows the transfer of knowledge from the pre-trained model to another dataset (*red*). In order to demonstrate the validity of our framework, we independently obtained the partitioning layer by iteratively freezing the layers one by one ($l$) and retraining the subsequent layers ($|L| - l$). This procedure was the second step, and illustrated in the second row. We performed this step twice, in order to have baseline results for the original `float32` model as well as for the converted to `bfloat16`. The accuracy of each obtained model with varying numbers of frozen layers ($l$) was evaluated, and the partitioning layer was identified based on the same primary condition. The maximum number of bottom layers should be quantized before partitioning with the highest possible accuracy of the model.

Consequently, in parallel, we obtained the same layer from our mixed-precision partitioning algorithm. The third row reflects the main goal of this work, which is the identified partitioning layer $l_s$, and, as a result, the mixed-precision model capable of on-device TL. This model consists of the frozen and quantized layers before $l_s$, and the subsequent layers (in `bfloat16`) for retraining. It is possible to keep the mixed-precision quantization for the frozen layers, as our algorithm provides this as well. However, as a general example for the evaluation part, we converted all frozen layers to `int8`.

Once the partitioning scheme has been identified and the partitioned mixed-precision model has been created, the next step is to retrain the model on the device. Figure 2 shows this procedure. During the retraining of the `bfloat16` layers, there is still a need of the quantized $q$-bit layers.

There are two options to update an upper part of the model. The first one is to create a new training dataset by passing the whole data through the bottom part of the model, and saving the output. We consider this possibility less feasible for embedded devices with limited memory resources. The second option is to use the bottom layers during each training epoch to pass data through to the upper

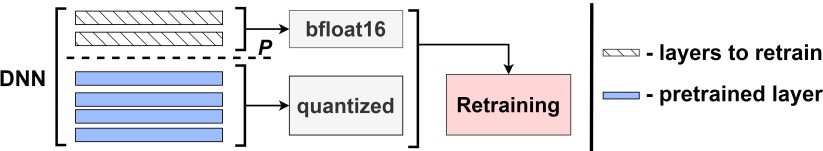

Figure 2: Retraining procedure of the partitioned mixed-precision model. The quantized bottom layers are only used to pass data through to the bottom layers.

layers. It is important to emphasize that the quantized bottom layers do not contain any additional backpropagation computations. The only contribution of these layers in the training procedure is to pass data through to the `bfloat16` layers.

## 5 RESULTS

In this section, we demonstrate the results of the described experimental setup, in order to validate our mixed-precision partitioning algorithm. Figure 3 shows the results of the proposed partitioning scheme for the ResNet-18 and SqueezeNetV1_1 models to perform TL locally for three datasets.

As described in subsection 4.2, in order to validate our approach, we identified the partitioning layer by iteratively freezing the layers and retraining the subsequent ones. *Blue* dots represent the retrained model in `float32`, while the *orange* ones represent the baseline `bfloat16` model. Each plot has a *red* vertical line that illustrates the partitioning layer to retrain the model. The preceding layers are frozen and `int8` quantized. The layers after this partition are in `bfloat16`. The green dots show the test accuracy of the mixed-precision models with this layer configuration.

The results yield the following conclusions. Firstly, all plots have an additional vertical line (*dotted, blue*) that indicates the accuracy of the DNN if it was split before the classifier. In all cases, the accuracy of the model that was retrained at a feature extractor layer was higher than that of the model that was split before the classifier. This highlights the need to consider the upper layers of feature extractor as potential layers of adaptation for transfer learning tasks.

Secondly, all plots demonstrate that our proposed mixed-precision algorithm successfully identified the uppermost partitioning layer without losing accuracy compared to the baseline accuracy of models in `bfloat16`. Moreover, the partitioning scheme was found without the necessity of retraining the model on a number of times equivalent to the total number of layers in the model. Hence, our algorithm significantly reduces the computational overhead.

Finally, the partitioned mixed-precision model achieves the same model performance as the full `bfloat16` model on new datasets during the TL tasks. As a result, the approach presented in this work successfully identifies the partitioning layer using layer robustness analysis. We verified that a model with mostly all quantized layers can leverage the knowledge from the pre-trained model and transfer it to new datasets. We believe that our approach has the potential to contribute to the realization of on-device TL in embedded devices.

## 6 DISCUSSION

In this work, we have considered the image classification task as a TL example. The approach presented in this work and the obtained results can be potentially applied to another ML tasks. We also focused on a clear separation between the fully frozen or quantized bottom layers and the updated upper layers. As we demonstrated, it is sufficient for a model to successfully transfer the knowledge to new datasets by updating only a few upper layers, while the bottom can be represented as it would be for performing an inference. However, the efficiency of on-device TL can be further improved by combining other works in this area. For example, by sparsity updating the weights of the upper layers, as shown in the work of Lin et al. (2022).

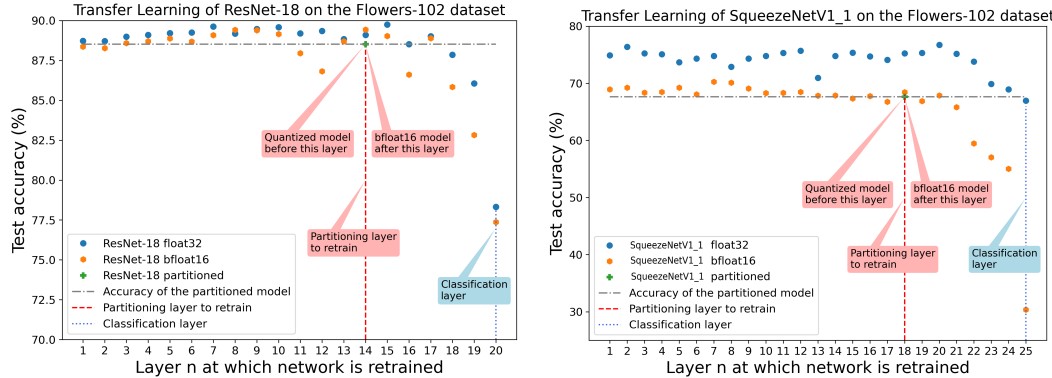

(a) TL of the ResNet-18 (*left*) and SqueezeNetV1_1 (*right*) models on the Flowers-102 dataset.

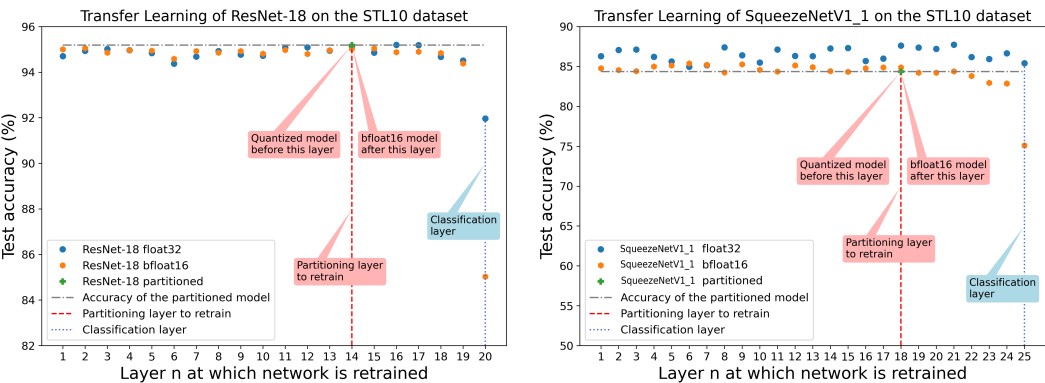

(b) TL of the ResNet-18 (*left*) and SqueezeNetV1_1 (*right*) models on the STL10 dataset.

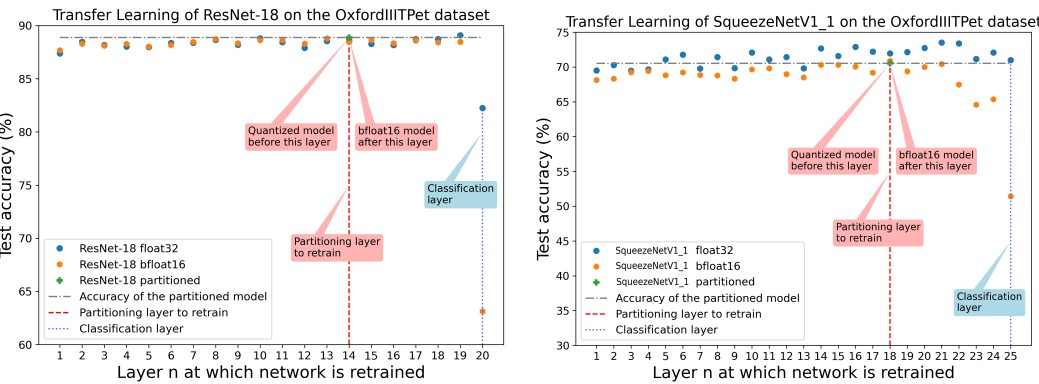

(c) TL of the ResNet-18 (*left*) and SqueezeNetV1_1 (*right*) models on the OxfordIIITPet dataset.

Figure 3: Applied Transfer learning to the ResNet-18 and SqueezeNetV1_1 models, including `float32` (*blue*), `bfloat16` (*orange*), and the mixed-precision *quantized* and `bfloat16` versions. The models were originally pre-trained on the ImageNet dataset, all new variations were trained on the Flowers-102 (*a*), STL-10 (*b*), OxfordIIITPet (*c*) datasets.

## 7 RELATED WORK

It is quite common practice in TL on edge devices to freeze feature extractor layers and train only classifier layers, often using only dense layers of the classifier part (Chiang et al., 2023), (Reguero et al., 2025), (Kang et al., 2024), (Valery et al., 2018). These approaches allow efficient TL, since only weights of the classifier are updated, which in turn requires less computational resources and

memory. However, the expressiveness of TL in this case is limited and the accuracy of the adopted model degrades, as it is demonstrated in section 2 or by Cai et al. (2020).

An alternative approach to TL on the edge that is superior to the aforementioned methods is to gradually freeze layers based on the per-layer convergence (Li et al., 2024), (Wang et al., 2023). The aforementioned methods determine the layers to be frozen during runtime, thereby reducing the time required for TL. However, in contrast to our approach, the initial model must fit into the device memory in order to perform the necessary inference and backpropagation at the early training stages. Consequently, since our approach limits the number of layers in backpropagation from the outset, we can significantly reduce the memory footprint in comparison to such methods achieving similar results.

Finally, the idea of partial updates of weights and biases in the backpropagation pass has been introduced and explored by a few studies such as Lin et al. (2022). Similar to our approach, these methods allow to reduce memory requirements for on-device training, enabling TL on edge devices. As an example, Cai et al. (2020) freeze the memory-heavy modules (weights of the feature extractor) and only update memory-efficient modules (bias, lite residual, classifier head) during TL, regardless of the position number of the layer. This methodology achieves memory saving compared to fine-tuning the full network. In our work, we used the fully quantized layers along with fully `bfloat16` layers split by the partition. Nevertheless, a combination of these two approaches to the upper layers of a network will be considered in the future, as they are complementary to each other.

## 8    CONCLUSION

In this paper, we presented our mixed-precision partitioning approach for transfer learning of DNNs in embedded systems. We emphasize that partitioning the model before the classifier improves the model performance. The partitioning algorithm identifies the potential partitioning layer through a process of layer robustness analysis. In order to allow resource-constrained devices to train locally, the algorithm maximizes the number of quantized layers and the top-1 accuracy while minimizing the computational precision. Investigating the best trade-off, we identified the partitioning scheme for a model. It consists of the `int8` quantized bottom layers and the `bfloat16` upper layers. We demonstrated our approach on the TL example for the image classification task, using pre-trained models and three additional datasets. We showed that the mixed-precision model can be retrained without losing accuracy compared to the baseline accuracy of the models in `bfloat16`. This leads to the conclusion that the mixed-precision model is able to leverage the knowledge from the pre-trained model to new datasets, and retrain locally on a device. Overall, our work can improve the efficiency and performance of on-device transfer learning in embedded devices.

## A    APPENDIX

Table 2 shows the additional details of the training setup, which was presented in the paper. We used the Adam optimizer as the optimization algorithm in all cases.

Table 2: Details of the training setup.

| | **Model** | | | |
| | ResNet-18 | | SqueezeNetV1_1 | |
| **Datasets** | learning rate | epoch, # | learning rate | epoch, # |
| Flowers-102 | $5 \cdot 10^{-4}$ | 10 | $10^{-4}$ | $80^*$ |
| STL-10 | $10^{-4}$ | 10 | $10^{-4}$ | 10 |
| OxfordIIITPet | $10^{-4}$ | 10 | $10^{-4}$ | 20 |

$*$ - In addition, we used data augmentation.

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
