# OpenReview forum: "On-Device Transfer Learning based on Mixed Precision Partitioning"
_ICLR.cc/2025/Conference — ICLR 2025 Conference Withdrawn Submission_

### Official Review · Reviewer_tNXV · 2024-10-27

**Soundness:** 1
**Presentation:** 3
**Contribution:** 1
**Rating:** 3
**Confidence:** 5

**Summary:**

The paper explores the research question of how we can improve the efficiency of on-device training. The authors proposed a solution to leverage different computation precisions over different partitioning layers.

**Strengths:**

1. The paper clearly demonstrates the importance of conducting on-device transfer learning.

2. The idea of leveraging different precision on different layers is an innovation, especially considering the robustness of different layers.

**Weaknesses:**

1. The paper is of a certain novelty. However, the novelty is the lack of solid support where many definitions are limited. For example, the authors define the problem only over bfloat16. I believe the solution should work with other cases but do not understand why authors particularly use this precision.

2. The solution is quite limited where it can be only applied to models corresponding to authors definitions. With the current realm of AI and ML applications, it should apply to more application. Otherwise, the authors should list certain evidence to show that it is common to do transfer learning with a selected partition layer. On the other hand, to the best knowledge of the reviewer, PEFT and QLoRA are more popular solutions for fine-tuning.

3. Experiments are conducted at a very trivial level. Only ResNet and SqueezeNet are used. More complicated model structure should be explored, for example, ResNet152 at least. The reviewer expects to see other models such as ConvNeXt, MobileNet (since the research question is defined over the scope of IoT devices such as Raspberry Pi and mobile devices), or even transformers like ViT. These are not very advanced models today but are important models to evaluate your solution. I also expect to see comparisons over large models where you can compare with methods like QLoRA.

4. The experiments cannot support the claim that you are doing transfer learning. In the experiment, the transfer learning is image classification by transferring ImageNet to another dataset. More applications should be considered.

**Questions:**

It looks like the proposed method, training (fine-tuning) after a particular partition layer. I typically have two questions.
1. First, does this mean that the proposed solution can only be applied to DNNs with typical structures. For example, diffusion models and UNets cannot use the proposed method as it is non-trivial to find multiple partition points.
2. Additionally, how much benefit can be brought with such a solution, compared to more welcomed solutions recently such as parameter efficient fine-tuning, with QLoRA asan a particular example? Pure LoRA adds more memory consumption as you need to calculate gradients for both backbone and adapters. But LoRA works pretty well with quantization, e.g. 4bit, which can greatly save memory. Since this paper also focuses on precision optimization, what are the advantages?

3. Why do typically define on bfloat16? It looks very limited. How about a layer with full precision?

4. In primary areas, you select "meta learning, and lifelong learning". Why is this paper related to them?

---

### Official Review · Reviewer_4gFn · 2024-11-02

**Soundness:** 2
**Presentation:** 3
**Contribution:** 2
**Rating:** 3
**Confidence:** 3

**Summary:**

In this paper, the authors propose a mixed-precision partitioning framework for on-device DNN transfer learning. The int8 quantization for bottom layers and bfloat16 for upper layers are combined during partitioning. The quantization sensitivity of the specific layer is utilized to perform robustness analysis for identifying the potential partitioning layer. The partitioning method achieves a trade-off between training performence and energy-efficiency on edge devices.

**Strengths:**

1. The research motivation is clear.
2. The paper is well written and easy to follow.

**Weaknesses:**

1. The specific challenging problems or key innovations need to be further clarified even though the ultimate goal of the method is to improve efficiency and performance for on-device transfer learning. Using an iterative approach to identify the partitioning point may be a natural choice, and mixed-precision quantization is also a common technique in deployment.
2. Critical experimental details are missing. How about the experimental device or the embedded system configurations?
3. It is necessary to supplement with essential experiments. More experiments should be provided to demonstrate the actual computational or energy efficency improvements for the proposed method running on embedded device. Especially, the resources consumption of the proposed method on embedded devices. The relevant claims in this paper will be more reliable.
4. There is no comparison between the accuracy of other potential partitioning layers and the accuracy of the partitioning layer identified using the proposed algorithm. In Figure 3, only the accuracy results corresponding to the identified partitioning layer are provided (marked with green dots and a gray line).

**Questions:**

See weaknesses.

---

### Official Review · Reviewer_vJtY · 2024-11-02

**Soundness:** 2
**Presentation:** 1
**Contribution:** 2
**Rating:** 3
**Confidence:** 2

**Summary:**

This paper proposes a mixed-precision partitioning approach for pretrained deel neural network by maximizing the number of quantized layers with the help of robustness to improve the performance in transfer learning.

**Strengths:**

1. The paper attempts to addresse the challenge of computational constraints in transfer learning problems.
2. Several model architectures and datasets are used in the experiments.

**Weaknesses:**

1. The writing in the paper is poor. The main contribution appears incremental and weak. The primary problem addressed in the paper is not well formulated. The paper includes eight definitions, some of which are unnecessary.
2. The main methodology (section 3.4 ) is not clearly demonstrated.
3. The experiments are not comprehensive and sufficient.
4. The paper does not compare existing methods in the experiments.

**Questions:**

1. What is main problem you want to solve in this paper?
2. Can you explain the results in Figure 3 more clearly?

---

### Official Review · Reviewer_vbHj · 2024-11-03

**Soundness:** 1
**Presentation:** 1
**Contribution:** 1
**Rating:** 1
**Confidence:** 4

**Summary:**

The paper introduces a mixed-precision partitioning framework designed to reduce training overhead for on-device training in transfer learning settings. The proposed approach partitions a pretrained network into two segments for classification tasks: the initial p layers are quantized and frozen, while the remaining L−p layers are set to bfloat16 precision and remain trainable for the target task. The authors also present an algorithm to identify the optimal partition point p for balancing partitioning efficiency. Preliminary experiments in limited settings suggest that the proposed partitioning scheme achieves accuracy on par with full-precision training.

**Strengths:**

1. The paper’s motivation is well articulated, investigating the hypothesis that “partitioning the model before the classifier can improve accuracy” under limited memory and computational resources. And recent studies inspire the integer format and bfloat16 format. investigation.
2. A sensitivity analysis is conducted using “the robustness of each layer to quantization” as an indicator of layer importance.

**Weaknesses:**

1. The paper has a limited scope. It is based on a specific partition with a specific quantization, with a limited transfer learning setting for two architectures on small-scale datasets such as OxfordIIITPet, Flowers-102 and STL10. These limitations cannot be justified with a “on-device training” setting, since future device will be more and more capable.
2. Writing and Clarity:
    - introduction mentions “cloud-to-device communication”, which seems not relevant.
    - The claim “before which all layers of the network are frozen” could imply that previous works freeze all layers, which may be misleading.
    - Terms like “population” and “generation” on lines 208 and 212 could be clarified for readability.
    - Statements such as, “While the algorithm tries to maximize the number of layers quantized to low bit integer precision, it tries to maximize the accuracy” cam ne rephrased for clarity.
3. The combination of mixed precison (integer and bfloat16) is not motivated well in section 2, where prior works have only one precision setup, without using mixed precision.
4. The paper does not show accuracy results or hardware profiling on any real edge devices.
5. The paper could benefit from providing a more detailed analysis of the impact of each layer on accuracy, as well as comparisons with alternative methods, such as a analysis on the impact of each layer, or executing the “multitude of approaches” in (Peccia & Bringmann, 2024), “sensitivity analysis in (Dash et al., 2022).”
6. Even for on-device TL, the decision on the partitioning should be done beforehand (if not, any justification?) for a given pretrained model,  where too much computational overhead in the searching is not a good excuse.
7. While Definition 7 outlines a broader quantization approach allowing variable q-bit precision for each layer, the implementation here exclusively uses q=8, which should be clarified to avoid misunderstandings.
8. The content in the PRELIMINARIES section might be condensed to fit the expectations of a top ML conference.
9. Certain acronyms (e.g., ONNX, HPC) could be briefly explained for clarity.
10. While open sourcing is valuable, it may not be positioned as a technical contribution.

**Questions:**

1. The procedure outlined on page 4, line 208 is somewhat difficult to follow. Could this be made clearer, perhaps through pseudocode?
2. Could the authors elaborate on why “converting all frozen layers to int8” is considered a general case for mixed-precision quantization?

---

### Note · Authors · 2024-11-22

I have read and agree with the venue's withdrawal policy on behalf of myself and my co-authors.